# PROXIMAL VALIDATION PROTOCOL

## ABSTRACT

Modern machine learning algorithms are generally built upon a train/validation/test split protocol. In particular, with the absence of accessible testing sets in real-world ML development, how to split out a validation set becomes crucial for reliable model evaluation, selection and etc. Concretely, under a randomized splitting setup, the split ratio of the validation set generally acts as a vital meta-parameter; that is, with more data picked and used for validation, it would cost model performance due to the less training data, and vice versa. Unfortunately, this implies a vexing trade-off between performance enhancement against trustful model evaluation. However, to date, the research conducted on this line remains very few. We reason this could be due to a workflow gap between the academic and ML production which we may attribute to a form of technical debt of ML. In this article, we propose a novel scheme — dubbed Proximal Validation Protocol (PVP) — which is targeted to resolve this problem of validation set construction. Core to PVP is to assemble a *proximal set* as a substitution for the traditional validation set while avoiding the valuable data wasted by the training procedure. The construction of the proximal validation set is established with dense data augmentation followed by a novel distributional-consistent sampling algorithm. With extensive empirical findings, we prove that PVP works (much) better than all the other existing validation protocols on three data modalities (images, text and tabular data), demonstrating its feasibility towards ML production.

## 1 INTRODUCTION

Most, if not all, machine learning production and research are conducted based on a train/test/validation set split protocol. A machine learning engineer or scientist often first receives a labeled dataset and splits it into a training and validation set, respectively. The role of the validation set is critical when considering robust model evaluation, selection, hyper-parameter tuning, etc. Post to the validation protocol, the best model being picked would be fed to the testing protocol, where the testing set is generally not accessible during real-world ML development till this phase.

Notably, prior to splitting the labeled dataset, one needs to determine the split ratio of the validation set against the training set. This ratio can be very tricky: if fewer samples are picked up for the training protocol, the model validation can be less reliable. Contrarily, the larger validation set effectively shortens the training resources, which may lead to performance degradation. In current days, this ratio is often set based on the experience level of a human expert. This problem anchored at the split ratio can also be exhibited in more complex validation schemes like cross-validation.

Indeed, this problem of setting the (sub-)optimal validation set is often, or mostly, ignored by the academic cohorts in the community. To date, as we scrutinize the related literature, very few works have touched down on this line (Li et al., 2020; Moss et al., 2018; Joseph & Vakayil, 2021). In hindsight, a large portion of the standardized academic benchmarks have had a prefixed validation set split, such as ImageNet (Krizhevsky et al., 2012), COCO (Lin et al., 2014), and SST (Socher et al., 2013). Also, the testing set is often visible for the evaluation of the academic research. On the one hand, this prefixed validation setup has some merit. For instance, this effectively dedicates the ML research to the model towards innovated model architecture, optimization methods, new learning paradigms, etc. On the other hand, we argue this could attribute to the technical debt (Sculley et al., 2015) of ML. When considering ML production for real-world applications, we make the following statements: (i)-not many application tags along with adequate or large-scale data because

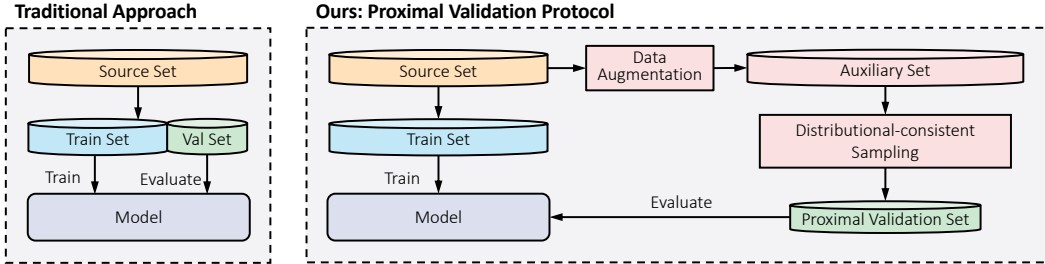

Figure 1: Traditional validation scheme (left) vs. Proximal Validation Protocol (right).

data curation and annotation both cost a fortune; (ii)-the testing set is often not accessible: considering the CTR prediction or manufactural defect detection where the testing data present and only present post to the deployment; (iii)-the validation set is almost always decided by the ML experts with their expertise. In these scenarios, how to split out a validation set may sit in the center. The benign condition — where a validation set is preset and fixed — almost always does not hold in ML production.

To this regard, we propose the **P**roximal **V**alidation **P**rotocol (dubbed PVP). With this novel validation protocol, we attempt to (fully) resolve the split problem and its trade-off. The core idea of PVP is rather simple. It first synthetically generates a validation set based on the labeled dataset without any splitting. Then, a novel distributional-consistent sampling algorithm is applied in order to select the most suitable synthetic data point for validation. The resulting set is dubbed the *proximal validation set*. Thanks to the proposition of the *proximal validation set*, PVP (in theory) does not rely on any real labeled data point for validation, effectively leading to performance improvement. Notable, the comparison of PVP with the conventional validation protocol is graphically depicted in Figure 1.

Empirically, we extensively conduct experiments on three data modalities — including tabular data, image data, and text data. We actively compare the PVP with standardized methods like the holdout protocol, K-fold cross validation, as well as the very limited related work like Joseph & Vakayil (2021). Besides the series of analytical justifications, we choose three major metrics to form a fair and comprehensive comparison: the *performance*, *t-v gap* and *variance*. Notably, *performance* means the test score (e.g., AUC and Accuracy) of a model, *variance* refers to the stability of the estimated performance (on validation set) under different random seeds, and *t-v gap* indicates the closeness of the estimated performance to the test one. We empirically show that PVP achieves better performance, lower bias and competitive variance than the standardized split-relied methods. With three major data modalities being experimented, we hope that the ideology and instantiation of the PVP can pave a way for a more effective validation protocol towards ML production on real-world applications.

At last, we may summarize the contribution of this work as follows:

- We propose a novel validation scheme-work — PVP — a stable and reliable validation protocol relying on only the synthetic data while capable of enhancing the model performance.

- The decent empirical results of PVP on three major data modalities manifest its "plug-and-play" nature. Its design is very much input data-dependent but independent of the model, architectures, optimizers, and tasks. We hope PVP can shed some light on data-saving, performance-effective, lightweight and profound validation procedure.

The code of PVP will be made public upon publication.

## 2 RELATED WORK

As we mentioned, the related literature remains very few. Looking back to the old days, the validation framework was raised to fix the issue of overfitting (Mosteller & Tukey, 1968; Stone, 1974; Geisser, 1975), which was first noticed by Larson (1931). Due to the universality of the data splitting heuristics, the split-relied method can be applied to almost any algorithm in almost any framework (Arlot & Celisse, 2010).

Recent works aim to reduce the validation estimate's instability, imprecision, and time cost. Specifically, the instability refers to variance between multiple training results with random seeds on the same model (Moss et al., 2018). And the imprecision means the gap between the model's evaluation results on the validation dataset and the test dataset (Zeng & Martinez, 2000). These works can be divided into two categories. Some works explore the variants of traditional data-split validation frameworks (Moss et al., 2018; Kohavi et al., 1995; Jiang & Wang, 2017; Székely & Rizzo, 2013; Jung, 2018; Li et al., 2020; Tiittanen et al., 2021; Zeng & Martinez, 2000), such as holdout and k-fold. The others try to propose a better split algorithm for the validation dataset generation. Joseph & Vakayil (2021) and Budka & Gabrys (2012) propose methods to sample a specific subset from the training set to generate a validation set in the tabular scenario. Joseph (2022) propose the optimal train/validate splitting ratio theoretically, but only for the linear regression model.

However, most, if not all, of the literature works are limited by a "split-relied" framework. Based on the logic chains from our previous section, they mostly would suffer from the splitting tradeoff problem. Among them, we deem PVP as the pioneer attempt to be fully split-free. The split-relied framework and suffer the train/validation split tradeoff.

## 3 METHOD

### 3.1 PROBLEM SETUP

To ease the discussion of validation methods, we start with problem setup. First, we define a *source set* $\mathbb{D}^{src}$ as a compilation of the training and potential validation set, i.e., an unsplit (labeled) training set. Similarly, the testing set is defined as $\mathbb{D}^{te}$. Further, we follow the well-known assumption that the instances of both datasets are I.I.D and drawn from an unknown underlying density distribution $\boldsymbol{F}(\mathcal{X}, \mathcal{Y})$:

$$\mathbb{D}^{src} = \{(\mathbf{x}_i^{src}, y_i^{src})\}_{i=1}^N, \ \mathbb{D}^{te} = \left\{\left(\mathbf{x}_i^{te}, y_i^{te}\right)\right\}_{i=1}^M, \ \text{where } (\mathbf{x}_i, y_i) \overset{\text{iid}}{\sim} \boldsymbol{F}(\mathcal{X}, \mathcal{Y}) \tag{1}$$

where $N, M$ denote the number of labeled instances in the *source set* and test set, respectively, and $\mathcal{X} \in \mathbb{R}^d$, $\mathcal{Y} \in \mathbb{R}$ are the input and label space. From the *source set* $\mathbb{D}^{src}$, we further define the validation and train set so as to formulate the traditional validation process (e.g., the holdout scheme).

**Definition 3.1 (Validation Process).** *We define a validation set, $\mathbb{D}^{val} = \mathsf{C}(\mathbb{D}^{src})$, where $\mathsf{C}$ is a (stochastic) collecting function facilitated by different seeds. This definition, in turn, yields a definition of a training set, $\mathbb{D}^{tr} = \mathbb{D}^{src} \setminus \mathbb{D}^{val}$, by eliminating the validation samples from the source set. From here, we further define a mapping function $f$, $f(\mathcal{X}) \to \mathcal{Y}$. Finally, we define the validation process as $\S(f, \mathbb{D}^{val})$, where $\S$ is a scoring function compiling the inference pass together with an evaluation metric, e.g., accuracy, f1-score.*

Now, we describe the evaluation metrics of the validation method, which is essential for judging the quality of the validation method. We formally declare the *T-V Gap* and *Variance* from the perspective of validation, following previous works (Arlot & Celisse, 2010; Zeng & Martinez, 2000).

**Definition 3.2 (T-V Gap).** *We define the test-validation gap (T-V Gap) as the difference between the test score(i.e., performance on the data space $(\mathcal{X}, \mathcal{Y})$) and the estimated score of practically yielded function:*

$$\left|\S(f, (\mathcal{X}, \mathcal{Y})) - \S(f, \mathbb{D}^{val})\right|. \tag{2}$$

**Definition 3.3 (Variance).** *We define $\mathsf{C}_i$ as a set of collection functions under different random initialization, and the variance is defined as:*

$$\frac{\sum_{i=1}^n \left(\S(f, \mathsf{C}_i(\mathbb{D}^{src})) - \overline{\S(f, \mathsf{C}(\mathbb{D}^{src}))}\right)^2}{n}, \tag{3}$$

Intuitively, *T-V Gap* signifies the inaccuracy of evaluation. A small *T-V Gap* corresponds to a precise estimate of the true score (usually refers to performance in the real-world test environment). The *Variance* measures the stability of the evaluation. And low *variance* means the estimated score is robust under collection function $\mathsf{C}$ with a different random seed. Ideally, with near-zero *T-V Gap*

and *Variance*, the evaluation result is reliable, stable and able to obtain the best function $f$ with the highest score (on the test set).

Usually, perhaps not practical on many occasions, when more labeled data is set for validation, i.e., gathered by $\mathsf{C}$ or $\mathsf{C}_i$, the greater intersection extent between the chosen set becomes larger. On the one hand, this would effectively reduce the *Variance*. Besides, this will make $\mathbb{D}^{val}$ closer to $(\mathcal{X}, \mathcal{Y})$, which would reduce the *T-V Gap*. On the other hand, this further causes a reduction of the labeled samples for training which may degrade the test score of the algorithm. Following this setup, in the pursuit of a decent validation set, we may term it a "split tradeoff".

Essentially, in the conventional randomized cross-validation or vanilla holdout schemes, it is always a plague to set the ratio between samples used for training and samples entertained for validation drawn from the *source set*. We find that this line of research is very much absent. Most research has set this split ratio as a preset meta-parameter and stuck with it throughout the development. However, as important as it is, we attempt to scrutinize this problem and propose a systematic "split-free" solution.

### 3.2 PROXIMAL VALIDATION PROTOCOL

#### 3.2.1 OVERVIEW OF FRAMEWORK

As introduced in definition 3.1, the validation set is obtained by a collecting function $\mathsf{C}$, i.e., $\mathbb{D}^{val} = \mathsf{C}(\mathbb{D}^{src})$, while the train set is formed via $\mathbb{D}^{tr} = \mathbb{D}^{src} \setminus \mathbb{D}^{val}$. However, to achieve a split-free solution, all data are left for training (i.e., $\mathbb{D}^{tr} = \mathbb{D}^{src}$). Thus, we design a new collecting function, termed $\mathsf{C}'$. The sample set yielded by $\mathsf{C}'$ is expected to have comparable evaluation quality to its traditional validation set counterpart — comparable *T-V Gap* and *Variance* (in definition 3.2 and 3.3), and comparable or superior performance.

As a result, we manage to build a framework PVP for $\mathsf{C}'$ and prove that it requires the following two critical components (colored with red in Figure 1):

- A data generator $\mathcal{G}$ to produce candidate samples from *source set* $\mathbb{D}^{src}$, which we define as *auxiliary set* $\mathbb{D}^{aux} = \mathcal{G}\left(\mathbb{D}^{src}\right)$. To tackle the obstacle of no available data for validation set construction when all labeled samples of $\mathbb{D}^{src}$ are saved for training, we generate a set of synthetic data. To prove this concept, we resort to the simplest method — the data augmentation family — to implement the data synthesis process. The reason is two fold: (i)-the augmented data can be arbitrarily large in theory, which may contribute to low *variance* by providing sufficient samples for validation; (ii)-the methods from the data augmentation family are generalized and can adapt to most of the data modalities and tasks.

- A *distributional-consistent sampling algorithm* $\mathscr{A}$ selects distributionally representative samples from the *auxiliary set* to form a validation set. The chief challenge lies in how to design the strategy to locate suitable samples among the *auxiliary set*, which can be chaotic and biased due to the randomization nature of data augmentation. Again, we propose a frustratingly simple method by relying on an angle-based distribution approximation (see Figure 2) to chase for a small *T-V Gap* evaluation.

In general, $\mathsf{C}'$ can be formulated as $\mathsf{C}' = \mathscr{A}\left(\mathcal{G}\left(\cdot\right)\right)$. And the output of $\mathsf{C}'(\mathbb{D}^{src})$, i.e., *proximal validation set* $\mathbb{D}^{pro}$, can replace the $\mathbb{D}^{val}$ in the validation process (in definition 3.1).

By proposing PVP, we entertain the possibility of relying (purely) on the generated samples to construct the validation process. Practically, this may pose a great number of advantages over the traditional split-relied validation process because it saves the samples for training, which likely leads to performance gains.

In the following, in correspondence to the aforementioned bullets, we detail the instantiation of the PVP framework.

#### 3.2.2 DATA GENERATOR

The first step to facilitating a proximal validation process is to form a candidate pool for the *proximal validation set*, which uses no original data points in the *source set*. We include an external data

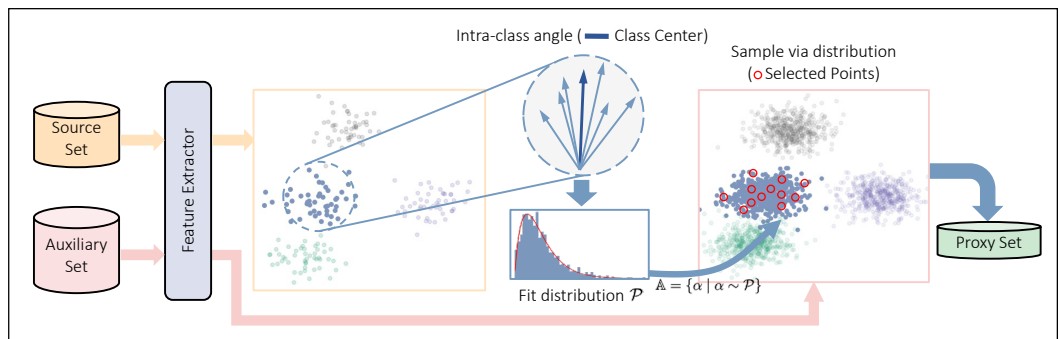

Figure 2: Illustration of distributional-consistent sampling algorithm. The steps of the algorithm are displayed from left to right. First, the distribution of $\mathbb{D}^{src}$ is estimated via an intra-class angular distribution on feature space. The estimated distribution is then used to select samples from the *auxiliary set* to form a *proximal validation set*.

generator $\mathcal{G}$ to conduct the construction. The function $\mathcal{G}$ can be implemented by many existing methods. Specifically, we profoundly choose the data augmentation approaches and carefully take them to adapt to the requirements of high validation quality in Section 3.1. Generally, we use this module to generate a large number of synthetic candidate data points, thanks to its continuous nature. By forming a large pool, we aim to bring down the *variance* metric during the final proximal validation. In addition, as is pointed out by Xu et al. (2022), the data augmentation scheme — when designed and implemented properly — is capable of covering the data space for the most part yet providing denser coverage. This may correspond to the metric of *T-V Gap* evaluation (definition 3.2) that relates to the following module.

Formally, we can define the data generator $\mathcal{G}$ and the candidate pool $\mathbb{D}^{aux}$ (named as the *auxiliary set*) more concretely as:

$$\mathbb{D}^{aux} = \mathcal{G}\left(\mathbb{D}^{src}\right) = \{g_0(\mathbb{D}^{src}), g_1(\mathbb{D}^{src}), \ldots, g_Q(\mathbb{D}^{src})\} \qquad (4)$$

where $\mathcal{G}$ consists of a set of $Q$ augmentation functions, i.e., $\mathcal{G} = \{g_0, g_1, \ldots, g_Q\}$. In particular, we conduct PVP on three fields, and the detailed augmentation methods are listed in Appendix A.1. There is admittedly rich literature around data augmentation; this is mostly embedded into the training stage for feeding more samples to train the model instead of for validation purposes.

Data generation is another feasible way to generate the *auxiliary set*. However, given the extendibility and computational advantages of the data augmentation methods, we stick PVP with them. We intend to leave the exploration of dataset generation methods, such as generative modeling (Goodfellow et al., 2014) and dataset distillation (Wang et al., 2018), to work further.

### 3.2.3 DISTRIBUTIONAL-CONSISTENT SAMPLING ALGORITHM

The prior data augmentation module can, in theory, produce an arbitrarily large number of samples for validation. This followed module is devised to select the most suitable samples to form the *proximal validation set* in place of the validation set of the conventional counterpart workflow. As mentioned in Section 3.1, high evaluation quality expects low *T-V Gap* and *variance*. It can be achieved if the distribution of $\mathbb{D}^{val}$ differs slightly from $\boldsymbol{F}(\mathcal{X}, \mathcal{Y})$ and $\mathbb{D}^{val}$ has a large volume. However, we do not have the precise form of $\boldsymbol{F}$. Instead, we only have $\mathbb{D}^{src}$, which is a set of realizations from $\boldsymbol{F}$. Thus, we can use the empirical distribution of the *source set* as a substitute for $\boldsymbol{F}$. And we abide by the expectation to propose the distributional-consistent sampling algorithm $\mathscr{A}$ to select sufficient samples that are distributional representations.

As mentioned in Section 3.2.2, while the augmented data may be able to wrap the data space of the source set, we must be careful with it due to the inductive bias from the data augmentation methods. Revealed from the literature (Zhang et al., 2015; He et al., 2019), the distribution of *auxiliary set*, produced by a set of augmentations, may drift from that of the *source set*. This drift, on the downside, may produce a large *T-V Gap* in the estimated score in the validation process if we directly employ the *auxiliary set* as a *proximal validation set*. Therefore, a sampling algorithm is indispensable in *proximal validation set* construction for less biased evaluation.

Briefly, we propose the simplest solution. This algorithm first characterizes the empirical distribution of *source set* $\mathbb{D}^{src}$ by an explicit angular distribution (Liu et al., 2020; 2017) and then samples

the angles via an explicit function to locate the corresponding points in the *auxiliary set* (as illustrated in Figure 2).

**Distribution Estimation with Angles.** The first step of our algorithm is to capture the empirical distribution of the *source set*. We adopt the simplest but most efficient method — intra-class angular distribution on feature space (Liu et al., 2020; Kobayashi, 2021; Liu et al., 2017) — for proof-of-concept of PVP. Unlike common collecting functions $\mathbb{C}$ that only consider inter-class distribution (e.g., stratified random sampling), our $\mathbb{C}'$ further measures intra-class distributions so as to control the distribution of validation set more finely, and ultimately keep the *T-V Gap* of the evaluation at a low level.

To be specific, we modeled the intra-class angular distribution on the angles between samples from $\mathbb{D}^{src}$ and their corresponding class centers. Given a sample $\mathbf{x}_i^y$ with category label $y$, let $\mathbf{z}_i^y = \Phi(\mathbf{x}_i^y)$ be the features extracted by an extractor $\Phi$, where we utilize BERT (Devlin et al., 2018) for text data and ResNet-18 He et al. (2016) for images. We define the calculation process of angle as $\mathcal{A}$ and define the angle of $\mathbf{x}_i^y$ relative to its class center as follow:

$$\alpha_i^y = \mathcal{A}(\Phi(\mathbf{x}_i^y)) = \arccos\langle \mathbf{z}_i^y, \mathbf{c}^y \rangle, \ \langle \mathbf{a}, \mathbf{b} \rangle = \frac{\mathbf{a} \cdot \mathbf{b}}{\|\mathbf{a}\| \cdot \|\mathbf{b}\|} \tag{5}$$

$$\mathbf{c}^y = \sum_{i=1}^{N_y} \frac{\exp(\mathbf{w}_i^y)}{\sum_{j=1}^{N_y} \exp \mathbf{w}_j^y} \mathbf{z}_i^y, \ \mathbf{w}_i^y = \frac{1}{N_y - 1} \sum_{i' \neq i}^{N_y} \langle \mathbf{z}_{i'}^y, \mathbf{z}_i^y \rangle \tag{6}$$

where $\mathbf{c}^y, N_y$ denotes the class center and the number of labeled instances of class $y$, respectively. Note that $\mathbf{c}^y$ is calculated by the weighted average of the features within the class $y$. Assuming the angles obey a Gaussian distribution, we can define the angular distribution for class $y$ as $\mathcal{P}_{(y)}(\alpha^y; \theta)$, and its parameters $\theta$ can be obtained via maximum likelihood estimation (MLE):

$$\hat{\theta} = \arg\min_{\theta} - \sum_{i=1}^{N_y} \log p\left(\alpha_i^y \mid \theta\right) \tag{7}$$

Notably, $\mathcal{P}_{(y)}$ is an explicit density function. To this end, we can obtain the empirical distribution of each class in $\mathbb{D}^{src}$.

**Sample through Distribution.** With the estimated distribution for the *source set*, our goal is to find representative points for the distribution as proximal validation samples. In particular, we conduct the process in a class-wise fashion. We generate angles via the explicit density function of the distribution to locate the corresponding samples in the *auxiliary set* for each class. Specifically, for a class $y$, we generate a set of angles by sampling from the distribution $\mathcal{P}_{(y)}$:

$$\mathbb{A}_y = \{\alpha \mid \alpha \sim \mathcal{P}_{(y)}\} \tag{8}$$

For each angle in $\mathbb{A}_y$, we search for samples in $\mathbb{D}^{aux}$ with the smallest angular gap to construct a *proximal validation set* $\mathbb{D}^{pro}$:

$$\mathbb{D}^{pro} = \{(\mathbf{x}, y) \mid \arg\min_{(\mathbf{x}, y) \in \mathbb{D}^{aux}} |\mathcal{A}(\Phi(\mathbf{x})) - \alpha|, \forall \alpha \in \mathbb{A}_y\}_{y=1}^{L} \tag{9}$$

where $L$ is the number of categories. At this point, we obtain a distributionally representative $\mathbb{D}^{pro}$ with sufficient data. It can be employed directly for validation while remaining all the data in $\mathbb{D}^{src}$ for training, as shown in Figure 1. And the entire process of PVP is shown in Algorithm 1.

## 4 EXPERIMENTAL EVALUATION

### 4.1 DATASET AND BASELINE

To validate our approaches, we experiment with three modalities of data: tabular, images, and natural language. All datasets are on classification tasks; notably, two are put in a long-tailed setup. The statistics of all datasets are listed in Appendix A.3. And these datasets contain different levels of data volume from 400-20k.

**Tabular Data.** We adopt 4 publicly available datasets from UCI (details in Appendix A.3), which are also data sources applied in other works (Wang et al., 2020; Bi & Zhang, 2018; Moss et al., 2018).

It is relatively easy to achieve stable and low-biased evaluations with balanced and sufficient data via cross-validation, which makes comparisons between methods less meaningful and persuasive. Therefore, a series of datasets with various imbalanced ratios are picked.

**Computer Vision.** The common and challenging long-tailed distribution is chosen as our evaluation environment. And we follow previous works (Park et al., 2021; Cui et al., 2019) to construct long-tailed versions of CIFAR10/100 with imbalance factor $\rho = 100$ and 10 respectively, named as *CIFAR10/100-LT*. The details about the construction are illustrated in section A.3.

**Natural Language Process.** We use the Reuters-21578 dataset (Dua & Graff, 2017) according to the exact setting in JKFold (Moss et al., 2018), where only corn and wheat categories are used.

**Model.** For tabular data, we use a decision-tree-based model xgboost (Chen & Guestrin, 2016) to perform classification tasks. While for CIFAR-LT and REUTERS, we train ResNet-44 (He et al., 2016) and DistilBert (Sanh et al., 2019) as classifiers, respectively. Notice that we suppress randomness in the model by fixing random seeds and maintaining a consistent batch data feeding order to enable the dataset (train/val set) to be the unique variable. And more implementation details are listed in Appendix A.4, including all training hyper-parameters.

**Method.** Generally, three baselines are considered to compare against. First, *holdout* with a train-val split ratio of 8:2. Despite it being widely used, especially in deep learning, owing to high efficiency, instability and deviation are its weak points due to the size of the validation set being still relatively small (e.g., near *few-shot* setup under hundreds of total samples). Second, *k-fold CV* where a choice 10 is taken for k. The instability can be improved versus *holdout*. Third, *repeated k-fold CV* (named as *J-K-Fold* (Moss et al., 2018)), the repeat times and k are set to 4 and 5, respectively. Theoretically, it is a further enhancement of stability at the cost of time. All these three methods are based on **stratified sampling**, which makes the percentages of different classes in both validation and train sets essentially the same. Additionally, we also add a customized method on the single tabular scenario as an extra baseline, i.e., *SPlit* (Joseph & Vakayil, 2021). *SPlit* is the latest splitting work, which utilizes support points (Mak & Joseph, 2018) to sample a specific subset from the train set as the validation set. And we set the sample ratio of *SPlit* the same as holdout (i.e., 8:2). Notably, the hyper-parameters mentioned above (i.e., 8:2 for *holdout*, 10 for *k-fold CV*, and 4-5 for *J-K-Fold*) are found by selecting the one with the best validation quality on the three metrics. And we run baselines and PVP **5** times for the image and text datasets considering time complexity issues, while **100** times for each tabular dataset.

**Metrics.** To comprehensively evaluate validation methods, we propose to compare three metrics simultaneously, i.e., *variance, T-V Gap, and scores on the test set*. (i) For *variance*, we calculate the standard deviation of estimated scores (scores on the validation set) over all runs. (ii) For *T-V Gap*, we use the mean of the absolute gap between scores on the validation set and the test set overall runs, which is the same usage in previous work (Zeng & Martinez, 2000; Budka & Gabrys, 2012). (iii) For *test score*, we use the mean of scores on the test set over all rounds. Likewise, the following combo — high *test score*, small *variance*, and *T-V Gap* — corresponds to good validation.

## 4.2 COMPARISON TO OTHER METHODS

We report our results in Table 1. We may conclude from the scores that: (i) The performances of models obtained by PVP are superior to all the rivals, and the improvement is significant on 4 datasets, i.e. BankMarket, PageBlocks, Diabetes and CIFAR10-LT. (ii) The *T-V Gap* are considerably reduced compared to all competitors, which is reduced by **32.3%** on average over the best baselines. (iii) The variance is maintained between *holdout* and *10-Fold CV*, and is closer to the latter, with higher evaluation stability.In a nutshell, PVP ameliorates both the *test score* and the *T-V Gap* at the same time while keeping the *variance* within a competitive level. These main results effectively justify the feasibility of PVP towards a proximal, split-free validation setup.

## 4.3 WHY DOES PVP IMPROVE MODEL PERFORMANCE?

Table 1 shows that the F1 score on all datasets is consistently improved compared to all baselines. We would like to attribute the performance improvement to the extra training data. Specifically, in our framework, all the given data (i.e., *source set*) can be used for training, without the necessity to

Table 1: Comparisons between PVP and other validation methods on seven classification datasets covering all three major data modalities, including tabular data(top), image(bottom left) and text(bottom right). The results are reported in F1, T-V Gap and variance, i.e. F1 $\uparrow_{\text{T-V Gap}\downarrow/\text{Var}\downarrow}(\uparrow/\downarrow$ indicates that the higher/lower the metric is, the better and vice versa. The numbers are scaled by $1e2$ for straightforward comparison.). And bold indicates superior results.

| METHOD | *BankMarket* | *PageBlocks* | *Diabetes* | *MushRoom* |
|---|---|---|---|---|
| *Holdout* | $59.1_{3.5/2.9}$ | $85.1_{3.3/4.0}$ | $72.5_{3.7/3.5}$ | $98.8_{0.8/1.2}$ |
| *10-Fold CV* | $59.4_{4.2/0.7}$ | $85.5_{4.7/1.1}$ | $72.7_{5.5/1.0}$ | $98.9_{1.4/0.2}$ |
| *4-5-Fold CV* | $59.1_{3.3/\mathbf{0.4}}$ | $85.9_{3.4/\mathbf{0.6}}$ | $72.8_{4.0/\mathbf{0.6}}$ | $98.9_{0.8/0.1}$ |
| *SPlit* | $59.6_{3.3/2.9}$ | $86.6_{2.6/2.9}$ | $73.5_{3.7/2.9}$ | $98.9_{0.7/0.8}$ |
| *PVP(ours)* | $\mathbf{61.3}_{\mathbf{2.6}/0.8}$ | $\mathbf{89.0}_{\mathbf{1.2}/1.3}$ | $\mathbf{77.4}_{\mathbf{3.1}/1.0}$ | $\mathbf{99.0}_{\mathbf{0.5}/0.6}$ |

| METHOD | *CIFAR10-LT* | *CIFAR100-LT* | METHOD | *REUTERS* |
|---|---|---|---|---|
| *Holdout* | $61.2_{8.4/2.3}$ | $47.0_{1.5/1.2}$ | *Holdout* | $88.7_{7.2/2.1}$ |
| *10-Fold CV* | $63.3_{7.7/0.6}$ | $49.9_{1.8/0.4}$ | *10-Fold CV* | $89.5_{7.1/0.5}$ |
| *4-5-Fold CV* | $61.1_{7.4/\mathbf{0.4}}$ | $47.4_{1.4/\mathbf{0.2}}$ | *4-5-Fold CV* | $88.9_{6.9/\mathbf{0.3}}$ |
| *PVP(ours)* | $\mathbf{67.3}_{\mathbf{1.8}/0.9}$ | $\mathbf{50.5}_{\mathbf{1.0}/0.8}$ | *PVP(ours)* | $\mathbf{90.0}_{\mathbf{6.8}/0.4}$ |

Table 2: Comparison of our PVP and other methods on distribution gap. Reported in Wasserstein distance over 10 times. (Tabular data(top), image(bottom left) and text(bottom right).)

| METHOD | *BankMarket* | *PageBlocks* | *Diabetes* | *MushRoom* |
|---|---|---|---|---|
| *Random(Holdout)* | $1.82 \pm 0.27$ | $9.54 \pm 1.32$ | $6.91 \pm 0.62$ | $2.46 \pm 0.37$ |
| *SPlit* | $1.75 \pm 0.20$ | $7.82 \pm 0.73$ | $5.87 \pm 0.63$ | $1.91 \pm 0.13$ |
| *PVP(ours)* | $\mathbf{1.56} \pm 0.16$ | $\mathbf{5.41} \pm 0.69$ | $\mathbf{3.90} \pm 0.72$ | $\mathbf{1.85} \pm 0.21$ |

| METHOD | *CIFAR10-LT* | *CIFAR100-LT* | METHOD | *REUTERS* |
|---|---|---|---|---|
| *Random(Holdout)* | $2.20 \pm 0.14$ | $1.85 \pm 0.03$ | *Random(Holdout)* | $3.02 \pm 0.37$ |
| *PVP(ours)* | $\mathbf{1.51} \pm 0.10$ | $\mathbf{1.57} \pm 0.03$ | *PVP(ours)* | $\mathbf{2.96} \pm 0.12$ |

partition some of them as the validation set. Owing to the split-free nature, PVP can save data for training to provide performance gains compared to the split-relied methods.

## 4.4 PROXIMAL VALIDATION SET VS TRADITIONAL VALIDATION SET

We compare *proximal validation set* with other validation sets in both quantitative and qualitative aspects: distribution gap and visualization difference. These two aspects demonstrate why the *proximal validation set* can work better than the traditional one.

**Proximal validation set has smaller distribution gap.** We compared the gap between the distribution of the validation set and the global distribution (approximated on *source set*) under different methods. The distribution is quantified via the intra-class angular distribution (as mentioned in Section 3.2.3), and the distance of two distribution is calculated by Wasserstein distance (Takatsu, 2008). As shown in Table 2, the distribution gaps of PVP are consistently smaller than the other methods. It empirically verifies that our validation set is a better representation of the global distribution, which can also lead to smaller *T-V Gap* in evaluation since $\mathbb{D}^{val}$ is more approximated to $\boldsymbol{F}$ in Eq 2.

**Proximal validation set are well spread out.** We visualize the 2D data representation produced by t-SNE in Figure 3. We use the MushRoom dataset, and different colors represent different ground-truth class labels. We contrast the TSNE embeddings of the validation set produced by two approaches: (a) *SPlit* (with split ratio 8:2), current best split-based methods, and (b) our method PVP. Compared with *SPlit*, we can observe that our validation points cover a wider region containing both central and marginal areas. Since PVP are not constrained by the split scheme, the number of validation samples in the *proximal validation set* can be larger and the coverage can be broader, which is presumed to lead to better evaluation quality.

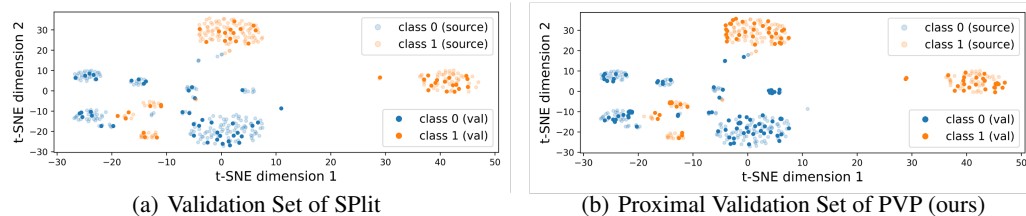

(a) Validation Set of SPlit    (b) Proximal Validation Set of PVP (ours)

Figure 3: TSNE visualization of the validation set representation on MushRoom. Different colors represent the different classes. The translucent points represent the *source set*, while the solid-colored ones represent the validation set. The details are referred to in Section 4.4.

Table 3: Using *proximal validation set* without extra training samples. The improvement of results was obtained by replacing the original split validation set with ours. The **-** implies deterioration, while the others are the default for improvement.

Table 4: Ablation Study. The deterioration of results obtained by using random sampling (from *auxiliary set*) instead of our distributional-consistent sampling. The **-** implies improvement while the others are default for deterioation.

| DATASET | T-V Gap(+) | Var(+) | F1(+) |
|---------|-----------|--------|-------|
| *BankMarket* | 1.39 | 1.40 | 0.06 |
| *PageBlocks* | 0.67 | 1.76 | 1.01 |
| *Diabetes* | 1.54 | 2.06 | 1.65 |
| *MushRoom* | 0.06 | 0.52 | 0.01 |
| *CIFAR10-LT* | 3.73 | 0.28 | 0.21 |
| *CIFAR100-LT* | 1.00 | 0.17 | 0.00 |
| *REUTERS* | -0.70 | 1.35 | 0.20 |

| DATASET | T-V Gap(-) | Var(-) | F1(-) |
|---------|-----------|--------|-------|
| *BankMarket* | 0.30 | 2.20 | 0.21 |
| *PageBlocks* | 3.50 | +0.12 | 0.05 |
| *Diabetes* | 3.35 | 0.22 | 1.23 |
| *MushRoom* | 0.06 | +0.09 | 0.00 |
| *CIFAR10-LT* | 22.87 | +0.05 | 1.23 |
| *CIFAR100-LT* | 21.01 | 0.06 | 0.44 |
| *REUTERS* | 5.16 | 0.33 | 2.30 |

## 4.5 USING PROXIMAL VALIDATION SET WITHOUT EXTRA TRAINING SAMPLES

We conduct experiments using the same training set as the *holdout*'s rather than the whole training set while replacing the original split validation set of the *holdout* with our *proximal validation set*. Table 3 shows that even without the extra training data, the performance can still be enhanced to some degree. This implies that the improvement in performance comes not only from additional training data but also from the high-quality validation set itself. Besides, we can also see that the *T-V Gap* of our method is still smaller in most cases when the validation set is the only different element. This comparison further confirms the superiority of our method in evaluation precision.

## 4.6 ABLATION STUDY

**Effects of Distributional-Consistent Sampling.** In this section, we attempt to reveal the effectiveness of the core algorithm in PVP. We compare PVP with a variant: *PVP-random*, which uses (stratified) random sampling from *auxiliary set* instead of distributional-consistent sampling. From Table 4, we can observe that the *T-V Gap* of *PVP-random* are aggravated, especially for the text and image datasets using deep models. Moreover, the test scores also degrade on most datasets. It signifies the indispensability of our sampling algorithm in the entire framework.

## 5 LIMITATION AND CONCLUSION

In this article, we proposed a Proximal Validation Protocol (PVP) to fully resolve the train/validation split trade-off. Through extensive experiments on datasets in different fields, including seven publicly datasets covering tabular, image and text data, we justify the comprehensive validity of PVP from both performance, robustness and reliability perspectives.

Indeed, we have chosen to leave the theoretical understanding of this framework out of the scope of this paper. Mostly, it is due to a general lack of theoretical study in this specific line of research around validation set construction. Rather, we intend to empirically prove this concept of split-free proximal validation with the simplest instantiation in this work. We hope to motivate the cohorts in the community to pay more attention to this research line because we believe it is vital and widely existing for ML production.

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

# A APPENDIX

## A.1 METHODS FOR GENERATING AUXILIARY SET

We list the detailed augmentation methods for each dataset in Table 5.

Table 5: The list of all the augmentation methods used for each dataset. $\mathcal{B}(a, b)$ denotes a beta distribution with parameters a and b. ***Times*** is the number of augmentation method performed per sample. (For Reuters, we utilize nlpaug[2] to implement these operations.)

| DATASET | Method | Parameters | Times |
|---|---|---|---|
| *BankMarket* [3] | MixUp | $\lambda \sim \mathcal{B}(0.5, 0.5)$:the mixing ratio | 30 |
| *Diabetes* [4] | MixUp | $\lambda \sim \mathcal{B}(0.5, 0.5)$:the mixing ratio | 30 |
| *PageBlocks* [5] | Smote | $N = 5$:the number of neighbors | 30 |
| *MushRoom* [6] | Smote | $N = 5$:the number of neighbors | 30 |
| *CIFAR-LT* | GridMask | d1=24, d2=33, rotate=1, ratio=0.4 | 5 |
| | Invert | - | 1 |
| | Solarize | $v \in \{256, 192, 128, 0\}$:solarization degree | 1 |
| | Brightness | $v \in \{1.9, 1.35, 0.95, 0.1\}$:an enhancing factor | 1 |
| *REUTERS* | ContextualWordEmbsSubstitute | $p \in \{0.3, 0.5\}$:augmentation proportion | 1 |
| | ContextualWordEmbsInsert | $p \in \{0.3, 0.5\}$:augmentation proportion | 1 |
| | SynonymAug | $p \in \{0.3, 0.5\}$:augmentation proportion | 1 |
| | RandomWordAugDelete | $p \in \{0.3, 0.5\}$:augmentation proportion | 1 |
| | RandomWordAugCrop | $p \in \{0.3, 0.5\}$:augmentation proportion | 1 |
| | RandomWordAugSwap | $p \in \{0.3, 0.5\}$:augmentation proportion | 1 |
| | BackTranslationAug | - | 1 |
| | AbstSummAug | - | 1 |
| | WordEmbsAugSubstitute | $p \in \{0.3, 0.5\}$:augmentation proportion | 1 |
| | WordEmbsAugInsert | $p \in \{0.3, 0.5\}$:augmentation proportion | 1 |

## A.2 PSEUDO CODE FOR PROXIMAL VALIDATION PROTOCOL

---
**Algorithm 1** Proximal Validation Protocol
---
**Input:** Source set $\mathbb{D}^{src}$, Number of category $L$, Classification model $f$, Scoring function $\S$, Feature extractor $\Phi$, Number of samples $m$, Data augmentation methods set $\{g_i\}_{i=1}^{Q}$

$\mathcal{G} = \{g_1, g_2, \ldots, g_Q\}$

$\mathbb{D}^{aux} = \mathcal{G}\left(\mathbb{D}^{src}\right)$

Initiate $\mathbb{D}^{pro} = \{\}$

**for** $l = 1$ **to** $L$ **do**

    Assume $p$ is a gaussian distribution

    $\hat{\theta} = \arg\min_\theta - \sum_{(\mathbf{x}_i, l) \in \mathbb{D}^{src}} \log p\left(\mathcal{A}(\Phi(\mathbf{x}_i)) \mid \theta\right)$

    $\mathbb{A}_l = \{\alpha_i \mid \alpha_i \sim p(\alpha \mid \hat{\theta})\}_{i=1}^{m}$

    **for** $\alpha_i \in \mathbb{A}_l$ **do**

        $\mathbb{D}^{pro} = \mathbb{D}^{pro} \cup \arg\min_{(\mathbf{x}, l) \in \mathbb{D}^{aux}} |\mathcal{A}(\Phi(\mathbf{x})) - \alpha_i|$

    **end for**

**end for**

Train $f$ on $\mathbb{D}^{src}$ to obtain $f^*$ with highest $\S(f, \mathbb{D}^{pro})$

**Output:** $f^*$, $\S(f^*, \mathbb{D}^{pro})$

---

[1] https://nlpaug.readthedocs.io/

[2] https://nlpaug.readthedocs.io/

[3] https://archive.ics.uci.edu/ml/datasets/bank+marketing

[4] https://archive.ics.uci.edu/ml/datasets/diabetes

[5] https://archive.ics.uci.edu/ml/datasets/Page+Blocks+Classification

[6] https://archive.ics.uci.edu/ml/datasets/mushroom

## A.3 DATASET STATISTICS

**Construction details of the long-tailed dataset.** As mentioned in Section 4.1, we construct long-tailed versions of CIFAR10/100 with imbalance factor $\rho$ = 100 and 10 following Park et al. (2021); Cui et al. (2019). Specifically, for a k-class dataset, we create a long-tailed dataset by reducing the number of examples per class according to the exponential function $n_i' = n_i \mu^i (\mu \in (0,1), i = 0, 1, ..., k)$, where $n_i$ is the original number of examples for class $i$, while $n_i'$ is the new number.

Table 6: Dataset statistics. $\rho$ denotes the imbalance factor, which is the ratio of the number of samples in the largest class to the smallest class. **#C** denotes the number of classes. **#F** denotes the number of feature dimensions. **#Source**, **#Test**, **#Auxiliary**, and **#Proximal(Ours)** denote the number of samples in *source set*, *test set*, *auxiliary set*, and our *proximal validation set*, respectively.

| DATASET | #Data | #C | $\rho$ | #F | #Source | #Test | #Auxiliary | #Proximal |
|---|---|---|---|---|---|---|---|---|
| *BankMarket* | 3047 | 2 | 7 | 13 | 2437 | 610 | 73110 | 2367 |
| *PageBlocks* | 5473 | 5 | 170 | 10 | 4378 | 1095 | 131340 | 3539 |
| *Diabetes* | 718 | 2 | 2 | 8 | 574 | 144 | 17220 | 1720 |
| *MushRoom* | 8124 | 2 | 1 | 22 | 406 | 7718 | 12180 | 207 |
| *CIFAR10-LT* | 22406 | 10 | 100 | 1024 | 12406 | 10000 | 210902 | 2980 |
| *CIFAR100-LT* | 29573 | 100 | 10 | 1024 | 19573 | 10000 | 332741 | 3000 |
| *REUTERS* | 520 | 2 | 2 | - | 393 | 127 | 5537 | 944 |

## A.4 IMPLEMENTATION DETAILS

**Tabular Data.** We used xgboost (Chen & Guestrin, 2016) as classifier for all tabular datasets. Specifically, we used the *XGBClassifier*[7] with parameters: objective of 'multi:softprob', booster of 'gbtree', early stopping rounds of 15, max depth of 4, learning rate of 0.1, n estimators of 50. While for the features of tabular data, we conduct some transformations on the original sample to generate features. The transformations contains two types: (i) for categorical feature, we use convert it to one-hot feature; (ii) for other types of features, we convert them to be normalized. And the dimension of feature varies among datasets.

**CIFAR10-LT and CIFAR100-LT.** We used ResNet44 from He et al. (2016) as the classification model for both CIFAR10-LT and CIFAR100-LT. We followed the same training procedure (train from scratch), initialization, and hyperparameters as He et al. (2016)(i.e., weight decay of 0.0001, momentum of 0.9, minibatch size of 128, total epochs of 200, initial learning rate of 0.1, learning rate decay stage of 100 and 150 epochs). While for the feature extractor, we used ResNet20 from He et al. (2016). And we train the extractor on *source set* for each dataset with same training hyperparameters as above. And the output of the last layer before classification is used as features, whose dimension is 72.

**Reuters.** We used DistilBert (Sanh et al., 2019) as classification model. We finetuned the pre-train model with training parameters: total epochs of 15, batch size of 8, weight decay of 0.01, learning rate of 2e-5. While for the feature extractor, we used Bert from Devlin et al. (2018). And we train the extractor on source set with same training hyperparameters as above. And the output in the 'cls' position of the last layer before classification is used as features, whose dimension is 144.

## A.5 ANALYSIS OF TIME COST

As shown in Table 7, we provided some quantitative measurements toward the running time of the *proximal validation set* construction process. The running time can be devided into two parts: (i) the pre-processing part, which only needs to be executed once, regardless of the number of training and validation process. This part contains the generation of *auxiliary set* (Section 3.2.2) and the estimation(Section 3.2.3). (ii) the sampling part, i.e., the second step in distributional-consistent sampling algorithm (Section 3.2.3), the execution number of which is consistent with training process. Indeed, the time cost of part one is high majorly due to the generation process.

---

[7]https://github.com/dmlc/xgboost

However, the operations in part one only need to be conduct only **once** no matter how many times we perform train and validation on the same dataset. While the operation included in **every** training process is the sampling part, which can be finished in a few seconds and therefore does not incur much additional time cost for the training process.

On the other hand, we position that the time cost of construction process can be attributed as a minor point and the reasons are two folds: (i)-this process can be faster via some engineering efforts like multithreading technology. (ii)-it can actually be viewed as a strategy to achieve a certain goal (better model performance and evaluation quality) at the expense of time, and such a strategy is common in many works, such as AutoAugment (Cubuk et al., 2019), RandAugment (Cubuk et al., 2020), FlipDA (Zhou et al., 2021).

Table 7: Time cost of PVP.(*s* denotes seconds and *m* denotes minute.)

| DATASET | ONE-TIME | | EVERY-TIME | |
|---|---|---|---|---|
| | *Generation(s/m)* | *Estimation(s/m)* | *Sampling(s)* | *Training(s/m)* |
| *BankMarket* | 65*s* | 13*s* | 3.0*s* | 4.0*s* |
| *PageBlocks* | 91*s* | 18*s* | 8.0*s* | 9.5*s* |
| *Diabetes* | 15*s* | 10*s* | 0.5*s* | 0.6*s* |
| *MushRoom* | 7*s* | 11*s* | 0.1*s* | 0.2*s* |
| *CIFAR10-LT* | 13*m* | 2*m* | 2.5*s* | 30.2*m* |
| *CIFAR100-LT* | 11*m* | 3*m* | 2.2*s* | 44.1*m* |
| *REUTERS* | 120*m* | 1*m* | 4.8*s* | 3.0*m* |

## A.6  A TEST CASE OF VALIDATION METHODS ON SYNTHETIC DATASETS.

Here, we test two standard validation methods on five synthetic datasets (created by sklearn.datasets.make_classification[8], train size of 1000, test size of 1000, class of 2 and features of 2) with increasing class imbalance in Figure 4. In this case, we can observe that these methods can work well in ideal scenarios with balanced data. In contrast, there is a risk of an imprecise estimation failure under unbalanced and small-scale scenarios, where the discrepancy between the test performance and the estimated one (i.e. the height of the yellow region) is undesirable large. This case illustrates that standard validation schemes do not always work perfectly even in a simple synthesis scenario, thus we can not easily rely on these methods in complex scenarios in the real world.

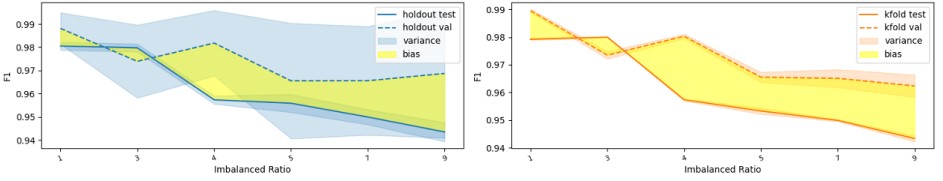

Figure 4: Holdout and 10 fold cross validation on synthetic dataset.

---

[8]https://scikit-learn.org/stable/modules/generated/sklearn.datasets.make_classification.html

