# OpenReview forum: "Proximal Validation Protocol"
_ICLR.cc/2023/Conference — Submitted to ICLR 2023_

### Official Review · Reviewer_cmsD · 2022-10-25

**Confidence:** 4
**Correctness:** 1
**Technical Novelty And Significance:** 2
**Empirical Novelty And Significance:** 1
**Recommendation:** 3

**Clarity, Quality, Novelty And Reproducibility:**

The paper is clearly written.

Quality of the evaluation is lacking. Test-Validation gap metric measures the difference of risk of validation and test set. One could, in principle, construct a validation set by merely corrupting the labels by the “right” amount and obtain a validation set that performs good on all the metrics considered. It would not mean that we should use this procedure as a procedure for constructing the validation set. Original construction of validation set is useful mainly because of the iid assumption of the data.

**Strength And Weaknesses:**

Strengths:

- Motivation for constructing new dataset is clearly presented.
- Method to construct a validation set is easy to understand

Weaknesses:

- Approach lacks theoretical basis. There is no reason to believe why the validation set is representative of the data generating distribution
- The metrics used for evaluation are not convincing to show the validation set is representative of test distribution

**Summary Of The Paper:**

Authors propose a new construction of validation set from training set without splitting the training data.  Authors construct the validation set in two steps:

- Data augmentation: Apply different data augmentation techniques, e.g. for image data, authors use GridMask, Invert, Solarize and Brightness, to create data generating procedure from the source dataset. Use this augmentation as a new data generator to consturct an auxiliary data source.
- Data selection from the auxiliary data: Authors select the data by choosing points that are close to the source data. Closeness is measured by using feature extractors, e.g. for image data authors use a pre-trained ResNet18 and text data authors use BERT.

Authors then empirically show that this validation procedure leads to higher accuracy and lower test-validation gap and smaller variance on different tasks.

**Summary Of The Review:**

I recommend rejecting the paper. I think there’s no reason to believe that the proximal set constructed is actually representative of the data generating distribution and thus should not be used as a validation set.

---

### Official Review · Reviewer_sKMx · 2022-10-29

**Confidence:** 4
**Correctness:** 3
**Technical Novelty And Significance:** 3
**Empirical Novelty And Significance:** 3
**Recommendation:** 5

**Clarity, Quality, Novelty And Reproducibility:**

### Clarity
- The address problem is clear and the method is also clear and simple.
- The paper is easy to read

### Novelty
- Make a validation set using data augmentation and angular-based sampling seems novel and interesting.

### Reproducibilty
- Althought the authors did not describe Reproducibility section, they submitted the source code.



**Strength And Weaknesses:**

### Strength
- The problem to be addressed is both practical and fundamental for ML community. When ML model is developed for real-world application in industry, it is quite challenging to make a validation set proper, fair, and effective.
- The proposed idea looks simple but effective using data augmentation and angular-based sampling.
- The experimental results present the efficacy of the proposed PVP against the conventional validation protocols in terms of three metrics.
- The paper is easy to read and understand.

### Weakness
I agree that this paper addresses a significant, practical and fundamental problem, so I think this paper can contribute to ML community. But, there are some room for improvements, in particular, of experiments.
- [Major] Considering that the method for proximal validation set leverages augmentation methods, classification tasks can benefit from this protocol. that is, multiple tasks such object detection, semantic segmentation in computer vision and QA in NLP might be diffult to be applied. I think the authors need to discuss or clarify this point.
- [Major] I think more experiments on industrial or real-world classification setups are required because the authors emphasize their motivation is from the debt of ML in industry. For example,
  - Most image recognition tasks leverage fine-tuning from pretrained models. Even if it is meaningful to the results on CIFAR-LT that the authors presented, more results on small-scale data such as Oxford-car/flower using ImageNet-pretrained will improve the contribution. Also, noisy label scenario experiments will be helpful.
  - Why did the authors use Distill-BERT instead of BERT for a classification model?
- [Major] More and intensive abation studies are requried for important hyperparameters to support their hypothesis because the authors do not present theoretical justification.
  - The size or sampling ratio of proximal valiation set from auxiliary set. It is not clear how to decide the size.
  - Because the authors argue that PVP is independent on model and optimizers, they need to present the evidence such as more experiments on one or two different models and an addtional optimizer.
- [Major] The authors should present statistical significance results in Table 1 and 2 at least, considering that this paper address evaluation protocol.
- [Minor] How is the performance under the below setup? This is also a alternative to address the lack of training data caused by validation set. Of coures, this cannot address the promblem of how to make a good validation set.
   - make a training (T) and valiation (V) set by spliting validation data  from source training set
   - make a new training set (T') merging training set (excluding valiation set) and the proximal validation set
   - Train T' and validate (V)
- [Minor] Tables 3 and 4 can be improved for better readability. For example, the authors use colored arrows (up-red, down-blue).



**Summary Of The Paper:**

This paper is a new validation protocol for classification tasks to address the fundamental problem of conventional training/valiation approaches by separating validation data from training data, called proximal validation protocol (PVP). For achieving this, the authors present a method to construct proximal valiation set by employing data augmentation and a sampling strategy basd on angular distribution.
To show the robustness of the PVP, they introduce two evaluation metrics: variance and T-V gap.
The authors validate their protocol on three data domains such as text, image, and tabular data, comparing with the standard valiation protocols such as hold-out and k-fold CV. The results seem meaningful.


**Summary Of The Review:**

Overall, I like this paper and agree the importance of the topics this paper addresses. However, some limitations exist in experiments.
So I decided my score as borderline . If the authors address my major concerns, I am willing to raise my score.

---

### Official Review · Reviewer_wMBi · 2022-10-31

**Confidence:** 3
**Correctness:** 2
**Technical Novelty And Significance:** 3
**Empirical Novelty And Significance:** 2
**Recommendation:** 3

**Clarity, Quality, Novelty And Reproducibility:**

The proposed strategy is somewhat novel to me and the results seem reproducible.

**Strength And Weaknesses:**

In general, this paper looks interesting to me. However, I have concerns about the empirical evaluation and theoretical guarantees.

Empirically, the proposed strategy is evaluated on several tabular datasets, image datasets, and one text dataset. However, most of these datasets are very small, which can not fully support the claims. For example, the authors use CIFAR10-LT and CIFAR-100-LT, I am wondering why the authors don't use the original CIFAR dataset or ImageNet. A similar issue also happens in the text data.

Theoretically, though the authors mention the theoretical analysis in the limitation. I still would like to see some theoretical results to support the claims in this paper, esp. when they only evaluate the proposed strategy on tiny datasets.



**Summary Of The Paper:**

In order to maximize the usage of the training set, this paper provides an alternative validation set construction strategy. Specifically, they construct an auxiliary dataset by augmenting the original training set. Then, they pick examples from the constructed auxiliary dataset. The new strategy is evaluated on several datasets.

**Summary Of The Review:**

This paper proposes and interesting validation set construction strategy, but the current version lacks enough empirical and theoretical supports.

---

### Official Review · Reviewer_iGVu · 2022-11-02

**Confidence:** 3
**Correctness:** 2
**Technical Novelty And Significance:** 2
**Empirical Novelty And Significance:** 2
**Recommendation:** 5

**Clarity, Quality, Novelty And Reproducibility:**

Clarity is discussed in weakness above. The propsed PVP idea appears original to me.

The paper is generally of decent quality, though the writing could be improved (suggestions below):

- Some quite unusual language is used for a scientific paper e.g.  'plague', 'entertained', 'frustratingly', 'profoundly'. I don't think this contributes much to the message of the paper and would reword.
- 'combo' -> 'combination'
- deterioation -> deterioration
- 'The split-relied framework and suffer the train/validation split tradeoff': doesn't make sense to me.



**Strength And Weaknesses:**

Strengths:
1. The paper tackles a clear issue (how best to choose a validation set) which has received little attention in the ML literature.
2. The proposed scheme, using data augmentation to generate a validation set, appears to be novel and goes some way to solving the issue of 'split tradeoff' problem.

Weaknesses:
1. The paper seems to only use compare validation sets in terms of how well they evaluate the performance of a *fixed model* (the hyperparameters/architectures in section A.4 all appear to be fixed and pre-defined), whereas the purpose of validation sets for model evaluation in ML, to me, is to be able to select between different models/hyperparameters. I think it is very important for the contribution of this paper to demonstrate having better model evaluation via PVP's validation set does actually lead to better model selection e.g. better hyperparameter tuning. If this is already shown in the experiments then it should be made clearer/explicit.

2. More generally, the paper is not presented clearly, at least to me. This is particularly the case in the experiments section, which is particularly problematic given that the paper focuses on showing the empirical strengths of PVP, rather than theoretical justification. This makes it hard to give a proper evaluation of the paper's contributions.

To start with, the 'F1' score is not properly defined, there is some alluding to it being the (top-1?) test accuracy in definition 3.1, which is what I take the definition to be (correct me if wrong). In any case, it is not clear what the values in table 3 and 4 represent: the word 'deterioration' alludes (but not in a clear way, and should be properly defined in any case) to the *difference* in each metric between using a proximal validation set vs. standard validation set (in table 3), and using random sampling vs. distributional-consistent sampling (in table 4). The reason why this lack of clarity is confusing is that if I understand correctly, there should be no difference between the F1 scores (test accuracies) in the settings presented in tables 3/4. This is because the choice of validation set shouldn't affect the test performance (for a fixed model trained on the same training data, see point 1 above), and hence that column should read as 0.

3. I'd also like to see more thorough empirical evaluation, e.g. to what extent does PVP depend on the choice of data augmentation. What happens to performance if we do just say additive Gaussian noise as our data augmentation? There are many choices that one can make for the data augmentation: is the empirical performance of PVP robust to this choice? Likewise, there seems to be a non-standard softmax-like weighting using for the class center in equation 6, is there a reason for this? Can the authors provide an ablation to this choice of weighting (say to just a standard uniform weighting)?

4. PVP seems to be designed for classification settings (particularly the distribution-preserving sampling), does it also extend to other settings say regression, or non-supervised learning regimes like self-supervised or unsupervised?



**Summary Of The Paper:**

This paper identifies the problem of 'split tradeoff', where common train/validation splits for data result in either less data to train on or poorer model evaluation. Instead, the paper introduces Proximal Validation Protocol (PVP), which, instead of splitting the validation set from a train set, creates a validation set through a combination of: 1) data augmentation on the train data and 2) a devised sampling scheme to preserve the distribution of the augmented validation set to be close to the train set. PVP allows one to use the whole training data in order to train models while still having a useful validation set for model evaluation, thereby avoiding the split tradeoff problem.

Experiments on three data modalities: tabular, image and natural language seem to demonstrate that PVP improves both the test performance of trained models, as well as reducing the gap between test and validation performance (i.e. validation is a better evaluation of model performance), whilst maintaining a competitive variance across different runs of their method (with different validation sets).

**Summary Of The Review:**

I am recommending weak reject. The problem identified is important and the proposed solution appears novel, but the experiments are both unclear and (as far as I understand) lacking in thoroughness, as detailed in my weakness section. I'm happy to be corrected though.

---

### Decision · Program_Chairs · 2023-01-20

**Decision:**

Reject

**Justification For Why Not Higher Score:**

While there are few strengths pointed out by the reviewers, all of the reviewers saw weakness outweighs the strengths. No reviewers were supportive of accepting. Several reviewers questioned theoretical grounding of the method and also found empirical evaluation and evidence does not overcome lack of theoretical guarantee. Also utilization of proposed protocol over regular validation set does not seem to be convincing to the reviewers.

In the end, the authors did not answer the reviewer's question or concerns. Original evaluation of two reviewers recommending reject and remaining two reviewers evaluating marginally below the acceptance threshold did not change. The paper does not satisfy the bar for ICLR conference publication.


**Justification For Why Not Lower Score:**

N/A

**Metareview: Summary, Strengths And Weaknesses:**

The paper proposed a new construction of validation set from training set without splitting the training set by using data augmentation and selecting data based on closeness of feature extractor. Authors perform experiments to show this validation protocol leads to higher accuracy, lower test-validation gap and smaller variance on different tasks

Strength
- Proposes interesting validation set construction strategy (wMBi, iGVu)
- Problem under question is practical and fundamental to ML community (sKMx, iGVu)
- Proposed idea is simple and effective (sKMx)
- Paper is easy to read and understand (sKMx, cmsD)
  - Reviewer iGVu found presentation of experiments unclear

Weakness
 - not sufficient empirical evaluation (wMBi, sKMx, iGVu)
  - lack of industrial or real-world task evaluation despite the motivation
  - more thorough evaluation on the choices
  - no demonstration of model selection ability using PVP (iGVu)
  - lacks theoretical base and guarantees (wMBi, cmsD, sKMx)
  - generality beyond classification tasks (sKMx ,iGVu)
  - metrics used for evaluation is not convincing (cmsD)